# Multi-dimensional social relationships shape social attention in monkeys

Sainan Liu[1,2†], Jiepin Huang[2,3†], Suhao Chen[2,4,5†], Michael L Platt[6,7,8], Yan Yang[2,3*]

[1]Division of Life Sciences and Medicine, University of Science and Technology of China, Heifei, China; [2]State Key Laboratory of Cognitive Science and Mental Health, Institute of Biophysics, Chinese Academy of Sciences, Beijing, China; [3]University of Chinese Academy of Sciences, Beijing, China; [4]Institute of Artificial Intelligence, Hefei Comprehensive National Science Center, Hefei, China; [5]Institute of Advanced Technology, University of Science and Technology of China, Hefei, China; [6]Department of Psychology, School of Arts and Sciences, University of Pennsylvania, Philadelphia, United States; [7]Marketing Department, the Wharton School of Business, University of Pennsylvania, Philadelphia, United States; [8]Department of Neuroscience, Perelman School of Medicine, University of Pennsylvania, Philadelphia, United States

*For correspondence:
yyang@ibp.ac.cn

[†]These authors contributed equally to this work

Competing interest: The authors declare that no competing interests exist.

## eLife Assessment

This study examined how multidimensional social relationships influence social attention in rhesus macaques, linking individual and group-level behaviors to attentional processes. The findings that oxytocin altered social attention and its relationship to both social tendencies and dyadic relationships are **important**, as recent technological advances allow for the exploration of neuronal activities and mechanisms in free-moving macaques. This work is **convincing** and will be of interest to those studying the interplay between social dynamics and information processing in primates.

**Abstract** Social relationships guide individual behavior and ultimately shape the fabric of society. Primates exhibit particularly complex, differentiated, and multidimensional social relationships, which form interwoven social networks, reflecting both individual social tendencies and specific dyadic interactions. How the patterns of behavior that underlie these social relationships emerge from moment-to-moment patterns of social information processing remains unclear. Here, we assess social relationships among a group of four monkeys, focusing on aggression, grooming, and proximity. We show that individual differences in social attention vary with individual differences in patterns of general social tendencies and patterns of individual engagement with specific partners. Oxytocin administration altered social attention and its relationship to both social tendencies and dyadic relationships, particularly grooming and aggression. Our findings link the dynamics of visual information sampling to the dynamics of primate social networks.

## Introduction

Social relationships bear profound significance for both the development and well-being of individuals within a community (*Taborsky and Oliveira, 2012*) and the cohesion and long-term success of their groups (*Testard et al., 2021*; *Testard et al., 2024*). Primates, including humans, engage in complex social relationships characterized by a wide range of interactions that are both diverse and multifaceted. Proximity serves as a cornerstone of social interaction, with individuals often seeking

closeness to one another for communication, bonding, and safety (*Schutte and Light, 1978*; *Stop-czynski et al., 2018*). Grooming is another fundamental behavior for primates, facilitating hygiene maintenance, parasite removal, and the cultivation and maintenance of social bonds (*Henazi and Barrett, 1999*; *Spruijt et al., 1992*). By contrast with these affiliative interactions, aggression supports resource competition and the acquisition and maintenance of dominance rank (*Hobson, 2020*; *Funk-houser et al., 2018*). The interweaving of individual affiliative and agonistic interactions contributes to the formation of social networks (*Hobson et al., 2019*; *Page, 2015*). Social networks within primate groups can be characterized at multiple levels or scales (*Aureli and Schino, 2019*; *Hinde, 1976*; *Grueter et al., 2012*), including broader patterns of social tendencies (e.g. whether a monkey is aloof or highly interactive), as well as individual engagement with specific individuals in that group (e.g. allies, friends, or enemies) (*Hobson et al., 2019*; *Hinde, 1976*). Together these affiliative and agonistic behaviors, both within specific dyads and in the aggregate, create the multi-level tapestry of social relationships in primate groups.

Primates, including humans, are highly visual animals (*Tsao et al., 2003*), and actively sample their environments to acquire useful visual information to guide behavioral decisions (*Krajbich and Rangel, 2011*; *Deaner et al., 2005a*; *Chang et al., 2013*; *Chang et al., 2012*; *Chang et al., 2015*; *Platt, 2002*). From birth, both human and nonhuman primates prioritize acquisition of visual social infor-mation, orienting preferentially to faces (*Deaner et al., 2005a*; *Heywood and Cowey, 1992*), and later during development using the gaze direction of others as a cue to important information in the environment (*Shepherd and Platt, 2008*; *Rosati et al., 2016*). Primates, including humans, also selectively attend to visual cues associated with social dominance and mate quality (*Chance, 1967*; *Deaner et al., 2005b*; *Shepherd et al., 2006*), as well as emotional states (*Eastwood et al., 2001*). Neural circuits supporting motivation and attention are intrinsically sensitive to visual social cues to dominance (*Klein et al., 2009*; *Watson and Platt, 2012*), potential mate quality (*Yorzinski and Platt, 2010*; *Klein et al., 2008*; *Watson et al., 2012*; *Adams et al., 2012*; *Smith et al., 2010*), and social dynamics (*Adams et al., 2021*). These observations endorse the idea that the primate brain is 'tuned' to support acquisition of visual information to guide adaptive social behavior.

Recent research highlights associations between social network dynamics and social attention. For example, patterns of grooming relationships predict patterns of mutual gaze in macaques (*Ball-esta and Duhamel, 2015*). Similarly, high-status monkeys selectively follow the gaze of other high-status monkeys whereas low-status monkeys follow the gaze of all monkeys (*Shepherd et al., 2006*). These findings suggest more complex, nuanced, and dynamic relationships between social networks and social attention in primates. A complete understanding of the dynamics of social attention, and its underlying biology, in human, or nonhuman primates, however, will require studying the visual orienting behavior of individuals living in groups whose social network dynamics have been quantita-tively characterized.

In this study, we evaluated social network dynamics in a group of four laboratory macaques that were raised together for several years, based on their patterns of general social tendencies and specific dyadic relationships in proximity, grooming, and aggression. We then removed individuals from the colony to assess basic visual orienting behavior while distracted by images of monkey faces of individuals both within and outside the social group, which were displayed on a computer monitor. Here, we define 'social engagement' as the broader patterns of interaction through which individ-uals express their social tendencies, observable in their interactions with all other members within the group (*Bentzur et al., 2021*). By contrast, we define 'individual engagement' as the interactions between specific pairs of individuals within the group (*Aureli and Schino, 2019*; *Bowler et al., 2012*). We examined correlations between patterns of social attention and patterns of overall social engage-ment with the group, as well as individual engagement with specific partners. This approach offers a quantitative assessment of how these social network dynamics shape patterns of social attention. Overall, we found that patterns of overall social engagement, based on aggression, grooming, and proximity, strongly shaped social attention, especially towards unfamiliar individuals. Furthermore, we found that exogenous administration of the social peptide hormone oxytocin (OT) blunted the impact of social engagement on social attention. By contrast, individual engagement in aggression within specific dyads shaped attention and OT administration increased the influence of aggressive relation-ships on social attention. Our findings reveal complex and nuanced interactions of multidimensional social relationships with social attention.

## Results

### An evaluation of multi-dimensional social relationships within a colony group

We assessed the social relationships of four rhesus macaques based on their spontaneous social interactions (*Figure 1a*). These four monkeys have resided in the same colony room for a minimum of 4 y. In their daily interactions, they exhibited a rich array of social behaviors, including proximity, grooming, and aggression. These behaviors serve important functions in communication, bonding, conflict resolution, and social organization within monkey facilities and natural habitats alike. These behaviors were measured automatically from video using a multi-stage detection based on YOLOv5 (*Figure 1—figure supplement 1*, see Methods for more details), and then evaluated mathematically (as an example dyad depicted in *Figure 1b and c*). We calculated interaction scores for pairs of monkeys using a dichotomous framework of 1 and –1 as in prior research (*Funkhouser et al., 2018*). Individual monkeys gained a social score of 1 for consistently displaying a tendency to approach others more frequently (proximity), receive more grooming from peers, and initiate aggression more frequently (shown by red arrows in *Figure 1d–f*).

We quantified each monkey's social engagement index (SEI) and individual engagement index (IEI). These models combine the scores of three dimensions of social relationships with independent weights. SEI summarizes the general social tendencies of each monkey based on their overall interactions with the other three individuals in the group. IEI quantifies the interaction tendencies within specific dyadic relationships. For simplicity, we first set all weights to 1 for both SEI (*Figure 1h*) and IEI (*Figure 1i*). This can be done by summing all interaction scores, as shown in *Figure 1g* with arrows indicating from individual with a higher score to one with a lower score. This approach yielded a quantitative account of social interactions, considering both the overall relationships and specific dyadic partnerships. For example, as depicted in *Figure 1f*, Monkey A exhibited stronger social cohesion with Monkey K, as measured by higher frequency of proximity. Nevertheless, when aggregating the proximity scores from all partners (Monkey K, L, and C), Monkey A had the weakest overall social cohesion, with a total proximity score of –1. Indeed, it was lower than Monkey K, who had a total score of 1 within the group. Three out of the four monkeys in the group (Monkey C, K, and A), as shown in *Figure 1i*, had individual engagement indices (colored symbols) that were not in line with their social engagement indices (gray open symbols). Hence, we use both SEI and IEI to understand the complexity of a social network. We next endeavored to probe how social attention varies with these two measures of multi-dimensional social relationships.

### Social engagement shapes attention towards monkeys' faces

We first probed whether monkeys with different SEIs respond differently in a classic social interference task (*Figure 2a*). In this task, a white square served as the visual target and the monkey had to maintain his gaze on it. A trial commenced with a fixation target appearing in the center of the screen for 800–1200 ms. In the social interference task (96% trials), an image (the distractor) was randomly presented eccentric either on the right or left side for 100 ms. After the first 50ms, a white gaze target was randomly illuminated either in the same hemifield as the distractor (congruent) or in the opposite hemifield (incongruent). Distractors were randomly selected from pictures of the faces of his three groupmates, pictures of the faces of three monkeys with whom he had no social interactions, or scrambled versions of these same face images. Each distractor had an equal chance of being presented (*Figure 2b*).

Example trials from an experimental session illustrate gaze shifts to the same target when different distractors were displayed in two different locations (*Figure 2c and d*). We calculated the response time to shift gaze to the target as our outcome measure of interest (see arrows in *Figure 2c and d*). We then focused analyses on response times as a function of distractor type and location within single sessions, to account for differences in motivation or attentiveness across days. Response times were faster when monkey faces were illuminated on the same (congruent) side as the target, compared to the scrambled images (*Figure 2e*, intact faces: 125.25±48.84ms; scrambled images: 142.80±47.87 ms; Student's *t*-test, $n_1$=382 trials, $n_2$=372 trials, p=7.87 × 10$^{-7}$). Conversely, response times were slower when the target was incongruent with display of a monkey face versus a scrambled version (*Figure 2f*, intact faces: 228.19±30.88 ms; scramble images: 213.97±29.96 ms; Student's *t*-test, $n_1$=373 trials, $n_2$=385 trials, p=2.17 × 10$^{-10}$). Distractor faces significantly impacted the behavioral performance of all

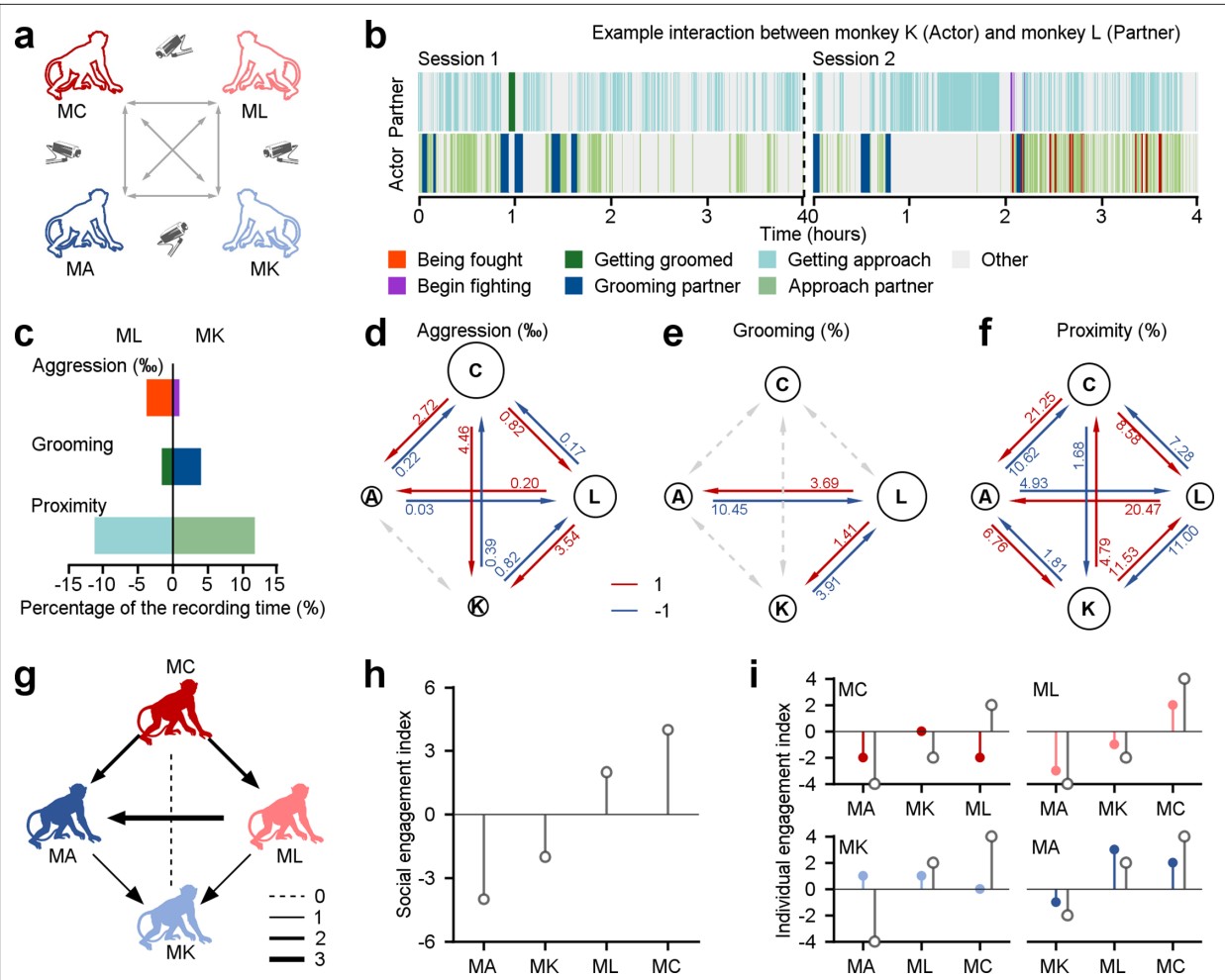

**Figure 1.** Different behavioral dimensions of social relationships among a group of four monkeys within a colony room. (**a**) The diagram depicts the procedure of acquiring social behaviors in various pairs of monkeys in the study. (**b**), Different social interactions exhibited by an example pair throughout the course of two experimental days. (**c**), The percentage of time engaged in three dimensions of behavior was quantified as a fraction of the total recording duration, in the given example pair. (**d-f**), Three dimensions of interaction behaviors in all dyads were evaluated across the total recording duration using a dichotomous framework with values of 1 and –1. The diameter of a given circle presents the cumulative scores obtained by that monkey in aggression (**d**), grooming (**e**), and proximity (**f**). The number on each arrow indicates the proportion of individuals engaged in the corresponding social behavior. (**g**), Social interactions among the group of four monkeys were quantitatively assessed based on three dimensions of social relationships, with an equal weight of 1 for each dimension. The thickness of lines corresponded to the interaction score in dyads. The monkey who receives an arrow is considered a negative score. Conversely, the monkey who sends an arrow earns a positive interaction score. (**h**), Social engagement index (SEI), which quantifies general social tendencies within the group, was computed by aggregating the scores of three social interactions between a specific monkey and the other three groupmates. (**i**), Individual engagement index (IEI; color-filled circles) evaluates the social interactions between a given subject and his specific counterparts, compared to their related SEIs (gray-empty circles).

The online version of this article includes the following source data and figure supplement(s) for figure 1:

**Source data 1.** Source data files for generating the results in *Figure 1*.

**Figure supplement 1.** A multi-stage automatic detection system designed to recognize social interaction between monkeys.

**Figure supplement 1—source data 1.** Source data files for generating the results in *Figure 1—figure supplement 1*.

---

four subjects, based on paired Student's *t*-tests across 72 conditions (6 intact faces ×12 experimental days) (*Figure 2g*, Congruent: MA: p=1.90 × 10⁻³⁵, MK: p=3.81 × 10⁻²⁷, ML: p=0.01, MC: p=4.37 × 10⁻¹⁸; Incongruent: MA: p=8.91 × 10⁻³⁴, MK: p=5.77 × 10⁻³², ML: p=8.66 × 10⁻⁹, MC: p=3.86 × 10⁻⁵).

We next calculated drifting time, which we defined as the difference between response times in the presence of intact faces and their scrambled versions (*Figure 2h*). Then we defined distractor bias as the difference between the drifting time in the incongruent and congruent conditions to quantify

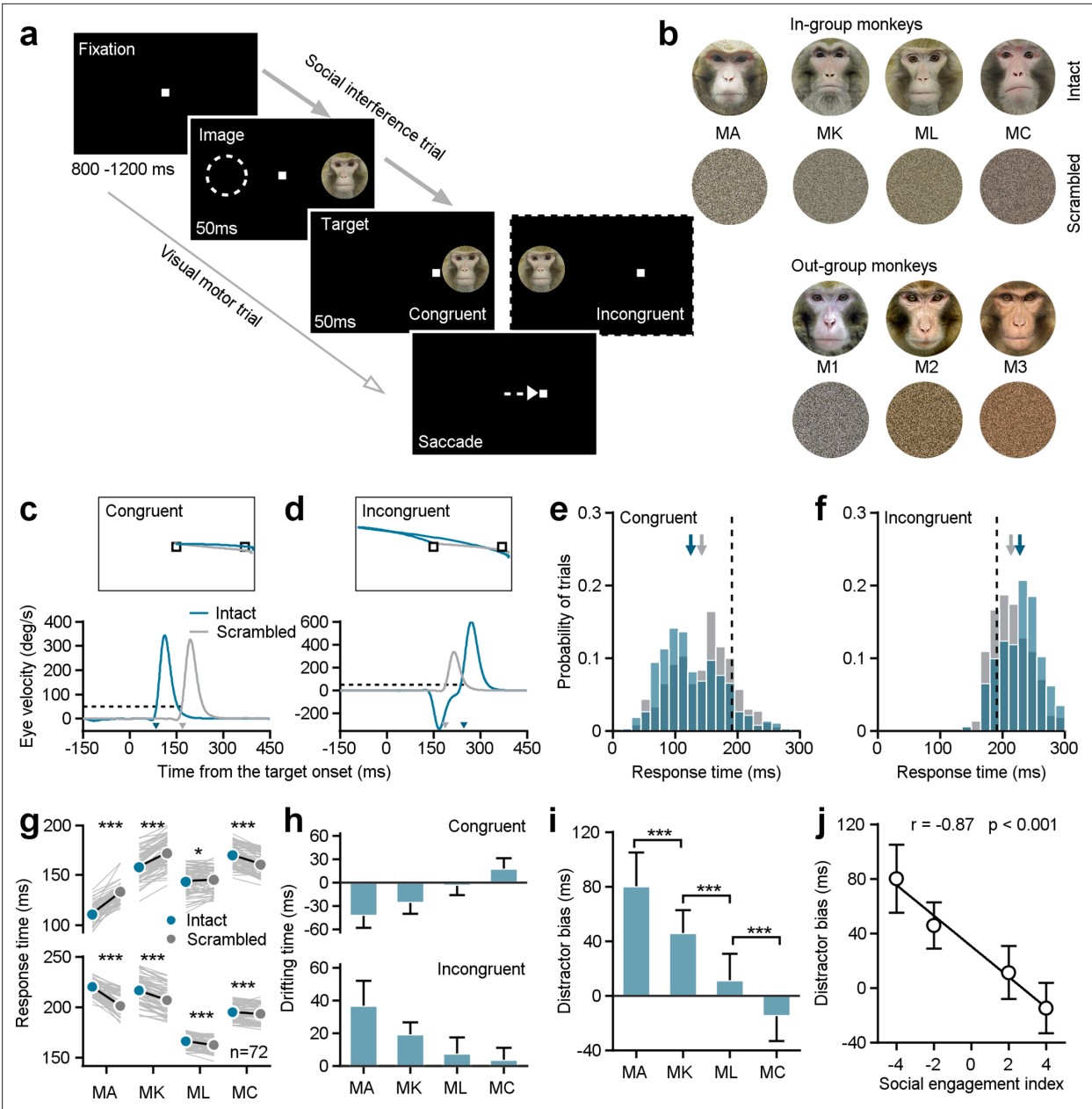

**Figure 2.** Individuals' social attention in response to the social interference task. (**a**) The experimental design encompasses both social interference trails and visual motor trials. (**b**) The distractor stimuli consist of seven monkeys' neutral face photographs and the related scrambled images. Four monkeys among them were raised in a colony group (in-group) and served as subjects in the study. The left three ones were from outside the group (out-group). (**c**, **d**) The eye position (top) and the horizontal eye velocity (bottom) on sample trials from Monkey A. The response times (arrows pointed) were determined utilizing a threshold of eye velocity at 50 deg/s and eye acceleration at 500°/s² (black dash lines). (**e**, **f**) The response time distributions were shown based on social interference trials from subject Monkey A's example experimental day, including data to an intact face picture (blue bars) and its scrambled image (gray bars) under congruent (**e**) and incongruent (**f**) conditions. The arrows present the corresponding average values, and the black dash lines show the average response time in visual motor trials. (**g**) The distractor of intact monkey faces had a notable impact on the saccade latencies of all four subjects. Each gray line represents the behavioral responses to a certain monkey face picture and its scrambled image on a given day. (**h**) The drifting time in the response time caused by the presence of both intact faces and their scrambled images. (**i**) To synthesize the interference effects of social attention on monkey intact face pictures, the difference in the drifting times between the incongruent and congruent conditions was analyzed as the distractor bias. The distractor bias varies significantly among the four subjects. Student's t-test, ***p<0.001; *p<0.05. MA and MK: p=2.32 × 10⁻¹⁷, MK and ML: p=6.77 × 10⁻²², ML and MC: p=7.73 × 10⁻¹⁴, Error bars: Mean ± SD. (**j**), Distractor biases of four monkeys in the group are negatively correlated with their SEIs$_{(1,1,1)}$ ($r$=–0.87, p<0.001), as determined by a two-sided Pearson's correlation coefficient. Error bars: Mean ± SD.

The online version of this article includes the following source data for figure 2:

**Source data 1.** Source data files for generating the results in *Figure 2*.

the impact of monkey face pictures on visual orienting—our measure of social attention (*Figure 2i*). Intriguingly, the distractor biases for individual subjects were negatively correlated with their social engagement indices ($SEI_{(1,1,1)}$) with an equal weight of one for all three social relationships (*Figure 2j*, two-sided Pearson's correlation coefficient, $r = -0.87$, $p = 4.51 \times 10^{-88}$). These findings indicate that individual variation in general social tendencies—from aggressive to prosocial—shapes social attention. Specifically, individuals with higher SEIs—that is those monkeys who were overall more social—showed a reduced tendency to be distracted by social information when performing a simple visual orienting task.

## General social tendencies differentially shape attention towards in-group and out-group monkey faces

In the wild, monkeys spontaneously differentiate between monkeys belonging to their own social group (in-group) and those belonging to other groups (out-group) (*Talbot et al., 2016*; *Pokorny and de Waal, 2009*). We next investigated the possibility that monkeys in our laboratory environment would exhibit similar behavior, and how this distinction would influence their social attention. We analyzed distractor bias according to whether the depicted monkey was in-group or out-group (*Figure 3a*). Before the study, subjects were familiarized with all distractor images on a daily basis for 8 wk, to rule out any novelty effects.

We compared response times to targets when either in-group or out-group monkey faces were displayed. On an example experimental day (*Figure 3b*), the subject monkey displayed slower response times in the congruent condition ($n_{in} = 195$ trials, $n_{out} = 193$ trials, Student's *t*-test, $p = 2.10 \times 10^{-3}$) and faster response times in the incongruent condition ($n_{in} = 188$ trials, $n_{out} = 186$ trials, Student's *t*-test, $p = 0.03$) when confronted with in-group monkey faces compared to out-group monkey faces. Next, we calculated distractor biases for the four subject monkeys when exposed to in-group and out-group monkey faces on individual experimental days. Distractor biases towards in-group monkey faces were significantly smaller than those towards out-group faces, across monkey subjects, on all experimental days ($n = 4$ monkey subjects ×12 experimental days, $p = 1.45 \times 10^{-7}$). We examined data from individual monkey subjects (*Figure 3c*, $n = 12$ experimental days, paired Student's *t*-test, MA: $p = 2.90 \times 10^{-3}$, MK: $p = 1.49 \times 10^{-2}$, ML: $p = 1.01 \times 10^{-3}$), with the exception of Monkey C, who gained the largest SEI with the highest social interaction frequency in the group (MC: $p = 0.14$).

We analyzed the probability of the first saccade being directed towards the distractor in the incongruent condition (*Figure 3—figure supplement 1a and b*). Monkeys showed a greater inclination to direct their initial saccades towards intact monkey faces (*Figure 3—figure supplement 1c*). Then, we calculated the difference in the probabilities of the first saccade that would be directed towards intact monkey faces and scrambled images. First saccades were more likely to be directed towards out-group than in-group monkeys relative to scrambled versions of the same images, for all monkey subjects on all experimental days (*Figure 3—figure supplement 1d*, $n = 4$ monkey subjects × 12 experimental days). Among them, Monkey A and L exhibited significantly higher first saccade probabilities towards out-group than in-group faces, relative to scrambled versions of the same images (*Figure 3—figure supplement 1e*, paired Student's t-test, $n_{in} = n_{out} = 12$, MA: $p = 5.28 \times 10^{-5}$, ML: $p = 2.41 \times 10^{-3}$; MK: $p = 0.54$, MC: $p = 0.44$).

Our analyses so far reveal that social interaction experience can blunt subjects' attentional capture by pictures of faces from monkeys within their group. Our experiments are limited by the fact that we only used 6 monkey faces as stimuli for both the in-group and out-group. Thus, the specific face pictures used might bias the observed differences between in-group and out-group faces (*Figure 3b and c*). To address this potential confound, we randomly withdrew a single image from an in-group monkey and one from an out-group monkey from the dataset, and reanalyzed distractor biases from the left two pictures of faces (*Figure 3—figure supplement 2a*). For each individual subject, this resulted in three sets of two distractors from the in-group or out-group monkey faces. Then, we have got $81(3^4)$ combinations from four monkey subjects, and finally a total of $81(3^4) \times 81(3^4)$ combinations of subjects and distractors from both in-group and out-group. Within each combination, we statistically measured the difference of distractor biases towards the left two pictures from in-group and out-group (*Figure 3—figure supplement 2b*). As depicted in *Figure 3—figure supplement 2c*, for 99.30% of the 6561 combinations distractor biases towards in-group monkey faces were significant less than those towards out-group faces (Two-sided Wilcoxon signed-rank test, $p < 0.05$). Thus,

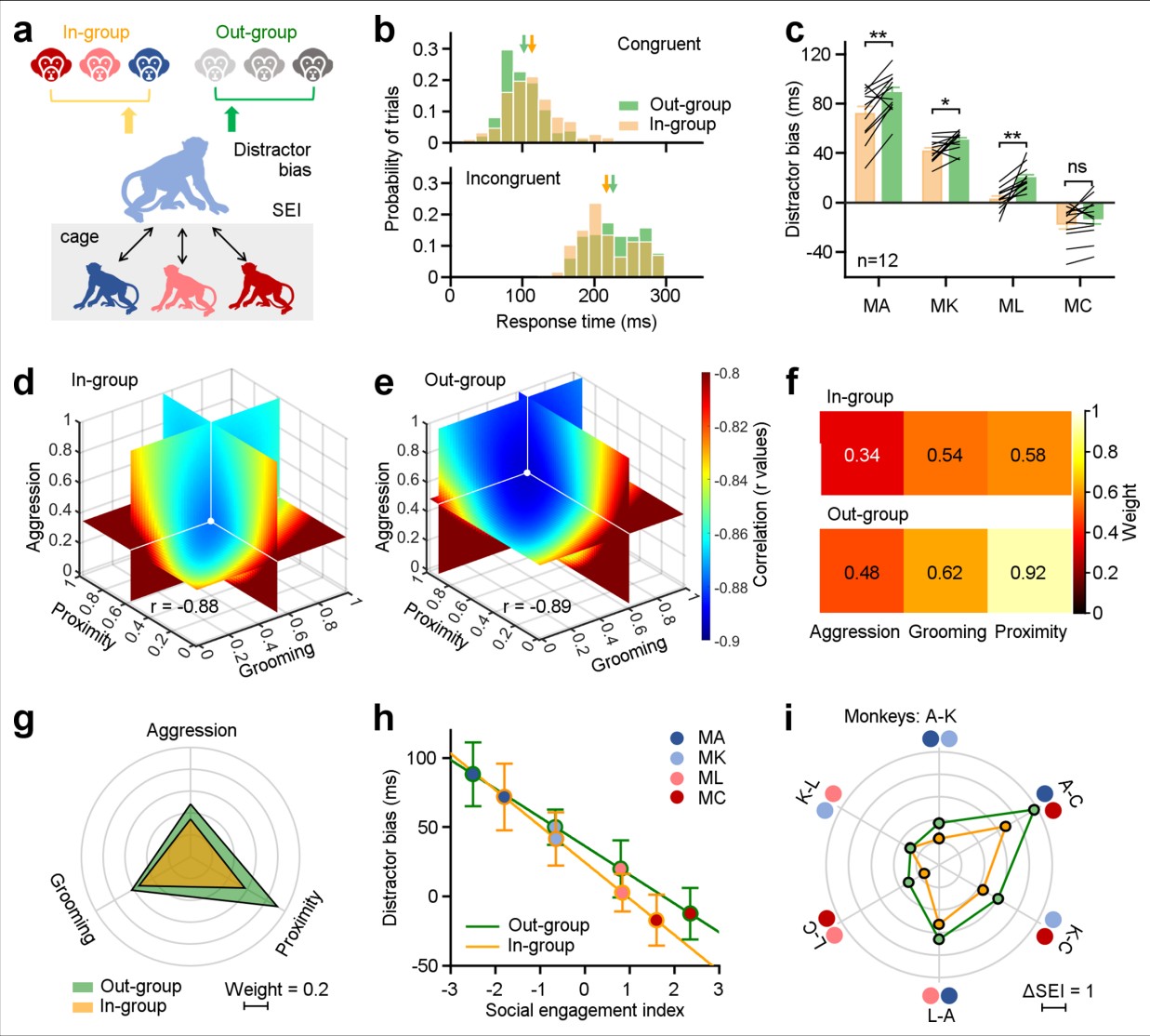

**Figure 3.** Three dimensions of social relationships distinctly shape attention towards in-group and out-group members. (**a**), Methodological schematic defining correlations between social engagement indexes (SEIs) and distractor biases towards the in-group and out-group members. (**b**) The distribution of distractor biases on an experimental day for the example Monkey A. Arrows indicate the related mean values (Congruent: In-group: 113.32±37.05 ms, Out-group: 102.21±33.34 ms; Incongruent: In-group: 216.94±40.26 ms, Out-group: 226.08±41.51 ms). (**c**) Individuals in the same group exhibited more social attentions in monkey faces from the out-group, except MC (paired Student's t-test, MA: p=2.90 × 10⁻³, MK: p=1.49 × 10⁻², ML: p=1.01 × 10⁻³, MC: p=0.14). Error bars: Mean ± SEM. (**d, e**) Quantification of weights for three behavioral dimensions computed from the correlations between SEI and distractor bias for in-group (**d**), and out-group monkeys (**e**). X, Y, and Z axes represent the weights of proximity, grooming, and aggression. The colormap illustrates the correlations between distractor bias and SEIs (r values), which were estimated by varying weights in the model from 0 to 1 with a step of 0.02. The white dot indicates the maximized correlation with the smallest negative r value. Each slice represents the r value surface along the X, Y, and Z axes, respectively. (**f, g**) Pseudo-colored heatmaps (**f**) and polar graph (**g**) show the 'best' weights for three social dimensions (in-group: $\omega_a$ = 0.34, $\omega_g$ = 0.54, $\omega_p$ = 0.58; out-group: $\omega_a$ = 0.48, $\omega_g$ = 0.62, $\omega_p$ = 0.92). (**h**) The liner relationships between monkeys' distractor biases and their SEIs with the 'best' weights. Error bars: Mean ± SD. (**i**) The differences in the SEIs between any two monkeys in H were scaled hexagonally in the circle, illustrating the magnitude of general social tendency in the colony. Compared to data towards out-group (green lines), the impact of social engagement was smaller when monkeys viewing in-group members (orange lines).

The online version of this article includes the following source data and figure supplement(s) for figure 3:

**Source data 1.** Source data files for generating the results in *Figure 3*.

**Source data 2.** Source data files for 3D colormap in *Figure 3*.

**Figure supplement 1.** Statistical analysis on the initial saccade directing towards the distractor in the incongruent condition.

**Figure supplement 1—source data 1.** Source data files for generating the results in *Figure 3—figure supplement 1*.

*Figure 3 continued on next page*

*Figure 3 continued*

**Figure supplement 2.** The effect of a single facial stimulus on distractor biases.

**Figure supplement 2—source data 1.** Source data files for generating the results in *Figure 3—figure supplement 2*.

**Figure supplement 3.** The correlation between distractor bias and three dimensions of social relationships in frameworks of social engagement and individual engagement.

**Figure supplement 3—source data 1.** Source data files for generating the results in *Figure 3—figure supplement 3*.

differences in social attention towards in-group and out-group monkeys are not likely driven by a specific picture in our stimulus set.

We next examined how general social tendencies of four monkey subjects varied with attentional capture by faces of in-group and out-group monkeys. We found that attention towards images of both in-group and out-group monkeys was highly correlated with the social engagement scores on three dimensions of social relationships: aggression, grooming and proximity (*Figure 3—figure supplement 3a–f*). The correlation coefficients varied across them, with aggression demonstrating the highest correlation with r values of about –0.8. Grooming exhibited a lower correlation compared to aggression, while proximity was the least correlated of the three.

We computed SEIs using a linear model with an independent weight setting for each dimension (see Methods for details), which summarized these three dimensions within the network of social engagement. We computed the best-fit weights to quantify how social attention varied towards in-group and out-group monkey faces as a function of subjects' social relationships (*Figure 3d–g*). The range of each $\omega$ was constrained to vary between 0 and 1, with a step size of 0.02. The correlations under different sets of $\omega_a$, $\omega_g$, and $\omega_p$ were calculated based on social attention towards in-group (*Figure 3d*) and out-group faces (*Figure 3e*). The best-fit weights were chosen by searching for the values of $\omega_f$, $\omega_g$, and $\omega_p$, that maximized the correlations (with smallest negative r values) between distractor biases and SEIs. As depicted in *Figure 3f and g*, the 'best' weights for aggression, grooming, and proximity ($\omega_a$, $\omega_g$, and $\omega_p$) were 0.34, 0.54, and 0.58, respectively, when monkeys viewed distractors drawn from in-group monkeys. The set of 'best' weights were 0.48, 0.62, and 0.92 when monkeys viewed distractors drawn from out-group monkeys. These weights of three dimensions for in-group data are smaller than those for out-group data. Comparing the weights for in-group and out-group distractors, the effect of proximity was larger than that of aggression and grooming. Then, the SEI was defined by these 'best' weights on three dimensions of social relationships.

We next examined the correlation between distractor biases and the SEIs determined by the corresponding weights. In *Figure 3h*, the yellow line indicates the correlation towards in-group with a slope of –26.22, whereas the green line indicates the correlation towards the out-group with a slope of –20.76. The difference of first saccade probability between intact and scrambled face pictures also showed a negative association with the SEIs (*Figure 3—figure supplement 3a-f*, two-sided Pearson's correlation coefficient, in-group: $r$=–0.71, p=1.21 × 10$^{-23}$; out-group: $r$=–0.73, p=1.12 × 10$^{-25}$). Differences in the SEIs between any two monkeys can serve as an indicator of the magnitude of general social tendencies within the group (*Figure 3i*). The impact of social engagement was smaller when monkeys viewed distractors from in-group (orange lines) compared to those from out-group monkeys (green lines). The average difference of the SEIs dropped significantly from 2.67±1.32–1.95±0.99 from out-group to in-group (paired Student's t-test, p=0.01). However, there was no relationship between the SEIs and performance in visual motor trials without a monkey face distractor, effectively ruling out any impacts of social tendencies on general oculomotor functions (*Figure 4—figure supplement 1d*, two-sided Pearson's correlation coefficient, in-group: p=0.08; out-group: p=0.18). Together, these findings indicate that behavioral tendencies that in aggregate define social networks dynamically shape monkeys' attention to social cues.

## Oxytocin blunts the impact of general social tendencies on social attention

Oxytocin (OT) is an endogenous peptide hormone that regulates crucial components of mammalian reproductive systems and influences social behavior more broadly (*Olff et al., 2013*; *Jiang and Platt, 2018a*; *Jiang and Platt, 2018b*). Here, we examined how oxytocin shapes the social attention of individuals as a function of general social tendencies. Monkeys were first acclimated to nebulized

exposure to saline for around 2 wk. During the experiment, monkeys were administered oxytocin (40 IU/mL, 2 mL; Beyotime) and saline by nebulization (Omron, infant nebulizer) for 2 min on alternating days. After a 15 min period of relaxation in a dim room, the monkeys participated in the visual orienting task.

We first examined the overall impacts of oxytocin on social attention by averaging distractor biases for all individual monkeys across all experimental days. *Figure 4a* shows that oxytocin significantly increased distractor bias by enhancing social attention towards out-group monkey faces (green bars, paired Student's *t*-test, n=48, p=0.01). Although there was a minor increase in distractor biases towards in-group distractors, this was not statistically significant across all individuals and experimental days (orange bars, p=0.21). We conducted additional analyses to investigate the impact of oxytocin on oculomotor function. We found no impacts on oculomotor behavior during visuo-motor trials without distractors (*Figure 4—figure supplement 1e*, n=48, p=0.31). There were also no significant effects on the probability of first orienting movement towards the distractor between saline and OT averaged across all monkeys (*Figure 3—figure supplement 1g*, n=48, paired Student's *t*-test, in-group: orange bars, p=0.93; out-group: green bars, p=0.41).

We next examined the impact of oxytocin on social attention as a function of general social tendencies. To do so, we analyzed the correlation between distractor biases and SEIs after administration of saline or OT (*Figure 4*), in a second set of experiments. The 'best' weights for distractor bias towards in-group and out-group following OT or saline are shown in *Figure 4b–e*. The impacts of OT were larger for in-group distractors compared with out-group distractors, with the weight of grooming showing the biggest effect (*Figure 4f*). We uncovered consistent and significant impacts of all 'best' weights on three dimensions of social behavior on social attention to both in-group faces (*Figure 4g*) and out-group faces (*Figure 4h*) (Two-sided Wilcoxon signed-rank test, $n_1 = n_2$=6, p=0.03). Following the administration of OT, we observed differences in the correlation between distractor bias and SEIs towards both in-group and out-group faces. The differences between saline and OT were greater for in-group faces compared to out-group faces (*Figure 4i and j*, two-sided Pearson's correlation coefficient, in-group: saline: p=$5.54 \times 10^{-45}$, *r*=–0.87, OT: p=$1.20 \times 10^{-47}$, *r*=–0.88; out-group: saline: p=$1.28 \times 10^{-40}$, *r*=–0.85, OT: p=$1.35 \times 10^{-41}$, *r*=–0.85; the slopes of in-group: saline: –26.07, OT: –60.47; the slopes of out-group: saline: –19.75, OT: –34.84). Subsequently, we compared the differences in the SEIs between any two subjects following administration of saline (empty circles in *Figure 4k*), the magnitude of general social tendencies was reduced following administration of OT (filled circles in *Figure 4k*). Thus, OT reduced the impacts of multi-dimensional social tendencies on social attention. Of the three measured dimensions, OT most strongly blunted the relationship between grooming—a prosocial, affiliative behavior—and attentional capture by both in-group and out-group faces.

## Social attention varies as a function of individual social relationships

We examined how individual pairwise interactions, rather than general social tendencies towards the entire group, vary with social attention. To do so, we calculated individual engagement indices which evaluated interactions within specific dyads as a function of proximity, grooming, and aggression (*Figure 5a*). Individual engagement scores varied with monkeys' social attention (*Figure 3—figure supplement 3g–i*), with aggression showing the highest correlation, followed by grooming, and then proximity.

We next estimated the best-fit weights for the correlations between distractor biases and IEIs, as detailed in the Methods and following the same procedure as for SEI analyses. Weights were chosen to maximize the correlations between distractor biases and IEIs (with largest positive r values, *Figure 5b*). We found that distractor bias and the IEI showed a strong linear correlation (*Figure 5d*, two-sided Pearson's correlation coefficient, p=$2.59 \times 10^{-21}$, *r*=0.69). Notably, there were large differences in the best-fit weights quantifying how SEIs and IEIs shape social attention towards in-group monkey faces. The 'best' weights for aggression, grooming, and proximity with SEIs were 0.34, 0.54, and 0.58 (*Figure 3g*). The 'best' weights of $\omega_a$, $\omega_g$, and $\omega_p$ with IEIs were 0.54, 0, and 0.04 (*Figure 5c*). Amongst all dimensions contributing to IEI, aggression was the most strongly associated with attentional capture by monkey face distractors.

We then assessed the impacts of oxytocin on social attention towards in-group members as a function of IEIs. To do so, we analyzed the correlations between IEIs and distractor biases towards in-group distractors following the administration of OT or saline. Two sets of 'best' weights were chosen by

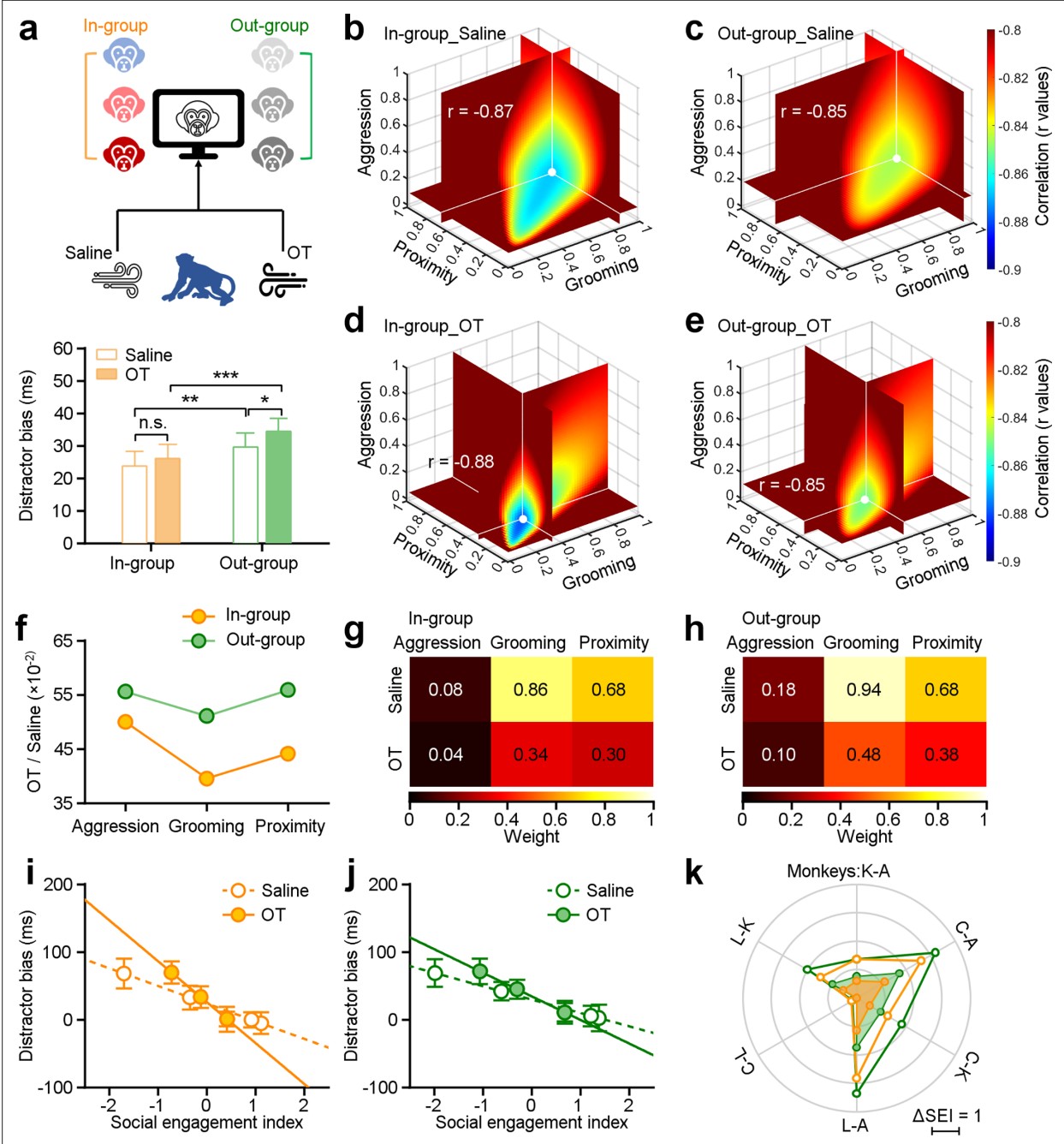

**Figure 4.** Impact of oxytocin on social attention regulation by multi-dimensional social relationships. (**a**), Experimental schematic (top) illustrating the administration of saline and oxytocin. Oxytocin greatly enhances the social attention towards distractors from outside the group (green bars, n=48), while there was no significant change in the group's attention towards faces from the in-group (orange bars) (*p<0.05; **p<0.01; ***p<0.001, paired Student's t-test. Error bars: Mean ± SEM). (**b-e**), The 'best' weights of three social dimensions by analyzing the correlations between social engagement indexes (SEIs) and distractor bias towards in-group and out-group monkey distractors following administration of saline (**b, c**) and oxytocin (OT) (**d, e**). (**f**), Change ratios of weights after OT and saline administration for in-group and out-group distractors. (**g, h**) Pseudo-colored heatmaps represent 'best' weights relating three social dimensions to in-group and out-group distractor bias under OT and saline. (**i, j**) Relationship between monkeys' SEIs defined by 'best' weights and their distractor biases towards in-group (**i**) and out-group (**j**) (the slopes of in-group: saline: –26.07, OT: –60.47; the slopes of out-group: saline: –19.75, OT: –34.84). Error bars: Mean ± SD. (**k**), The differences in the SEIs between any two monkeys following administration of saline and OT (**i, j**) were scaled hexagonally in the circle. Compared to data under saline (empty circles), OT (filled circles) reduced the impact of social engagement on social attention under both conditions: viewing in-group and out-group monkey faces.

The online version of this article includes the following source data and figure supplement(s) for figure 4:

*Figure 4 continued on next page*

*Figure 4 continued*

**Source data 1.** Source data files for generating the results in *Figure 4*.

**Source data 2.** Source data files for 3D colormap in *Figure 4*.

**Figure supplement 1.** Saccadic eye movements observed during visual motor trials were unrelated to social relationships within the group.

**Figure supplement 1—source data 1.** Source data files for generating the results in *Figure 4—figure supplement 1*.

maximizing the r values (saline: *r*=0.63, *Figure 5e*; OT: *r*=0.64, *Figure 5f*). Surprisingly, OT, compared to saline, doubled the weight for aggression (*Figure 5g*). After OT administration, we observed a decline in the slope between IEIs and distractor biases (*Figure 5h*, two-sided Pearson's correlation coefficient, saline: p=3.47 × 10$^{-17}$, *r*=0.63; OT: p=3.65 × 10$^{-18}$, *r*=0.64). Specifically, the slope for saline was 48.63, whereas the slope for OT was 23.21. These findings imply that OT amplifies the impact of dyadic aggressive interactions on social attention.

## Discussion

Social relationships emerge from the way individuals perceive and interact with the world around them. In this study, our goal was to probe links between social relationships among four monkeys in a group and behavioral performance in a separate computerized attention task. We specifically investigated how potentially distracting pictures of in-group and out-group monkeys impacted performance of a simple visual orienting task. We focused on three dimensions of social interactions (aggression, grooming, and proximity) at two distinct levels (general tendencies vs. dyadic relationships)

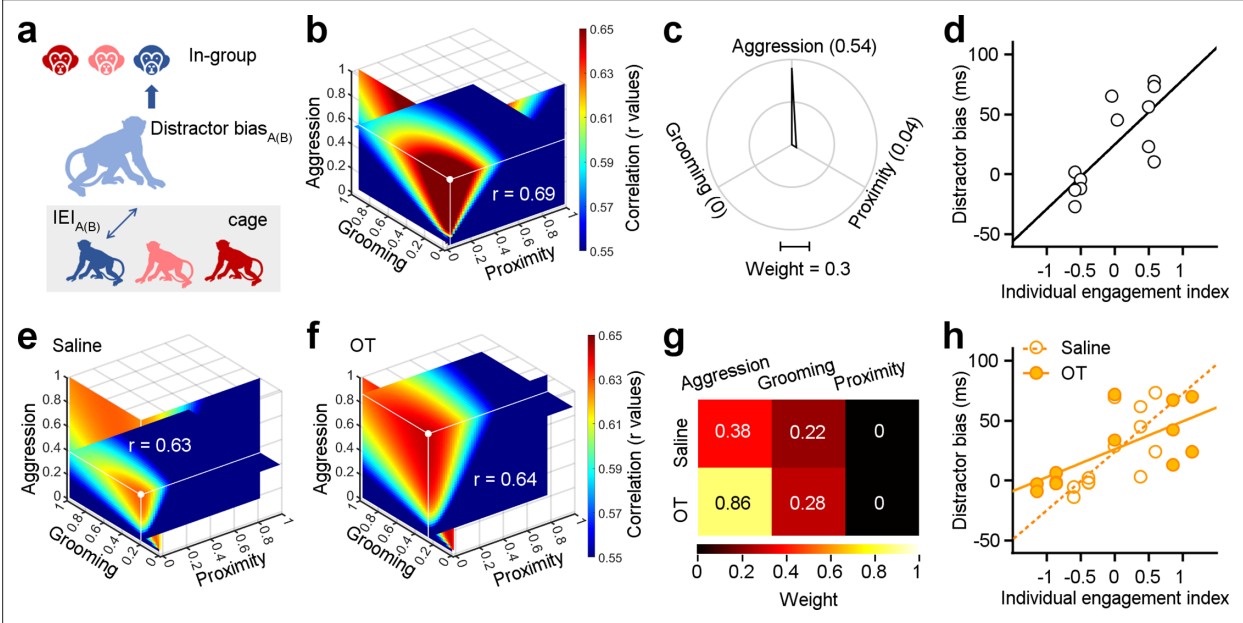

**Figure 5.** Individual engagements on three dimensions of social behavior distinctly shape social attention towards in-group. (**a**) Methodological schematic defining the correlations between the individual engagement index and the distractor bias towards in-group. (**b**) Quantification of weights for three behavioral dimensions computed from correlations between individual engagement indexes (IEIs) and distractor biases. The intersection point of the three slices in the graph signifies the position of the maximal r value (white point), with each slice representing the r value surface along the X, Y, and Z axes, respectively. (**c**) Polar graph shows the 'best' weights for three social dimensions ($\omega_a$ = 0.54, $\omega_g$ = 0.04, $\omega_p$ = 0). (**d**), Relationship between IEIs with 'best' weights and distractor biases (two-sided Pearson's correlation coefficient, p<0.001, *r*=0.69). (**e, f**) The 'best' weights of three social dimensions by analyzing the correlations between IEIs and distractor bias towards in-group following administration of saline (**e**) and OT (**f**). (**g**) Pseudo-colored heatmaps represent the 'best' weights of three dimensions of social interaction behavior in (**e**) and (**f**). (**h**) Relationship between IEIs and distractor biases (two-sided Pearson's correlation coefficient, saline: p<0.001, *r*=0.63; OT: p<0.001, *r*=0.64. The slope of saline: 48.63; the slope of OT: 23.21).

The online version of this article includes the following source data for figure 5:

**Source data 1.** Source data files for generating the results in *Figure 5*.

**Source data 2.** Source data files for 3D colormap in *Figure 5*.

and analyzed correlations with task performance. We mathematically assessed how social tendencies within the group, as well as specific dyadic relationships, varied with attentional capture by social stimuli. We also explored how the social peptide hormone oxytocin influenced the impact of multi-dimensional social relationships on social attention. Our findings offer insights into how social attention varies with both general social tendencies and specific dyadic relationships, thereby exposing links between visual social information processing and the dynamics of social networks in primate groups. Our study revealed that social attention is sensitive to group membership and reflects multi-dimensional social relationships (*Figure 3*). Social attention reflected both general social tendencies (SEIs, see *Figure 3g*) and individual dyadic relationships (IEIs, see *Figure 5c*), even for the same social cues (e.g. in-group). Our work provides a nuanced perspective on the mechanisms mediating the acquisition and use of social information.

Neural processes that distinguish in-group and out-group conspecifics presumably evolved because cooperation within groups conferred survival advantages (*Ten Velden et al., 2017*). Individuals who formed strong bonds within their group were better equipped to secure resources, protect against predators, and navigate social hierarchies (*Jablonski, 2021*). Consequently, abilities for recognizing and favoring in-group members, as well as kin, are thought to have evolved as adaptive strategies that maximize fitness and maintain social cohesion (*Bergman et al., 2003*). These adaptations can manifest as both in-group favoritism and out-group dislike (*Brewer, 1999*). Such biases foster solidarity and cooperation within a group, but can also fuel intergroup conflict, prejudice, and discrimination (*Halevy et al., 2008*). Here, we found that monkeys spontaneously differentiate members of their own social group (in-group) from those outside their group (out-group) when performing a simple visual orienting task. This distinction was evident in heightened vigilance towards the faces of out-group monkeys compared to those of in-group monkeys (*Figure 3c*). Moreover, this tendency varied among individuals. Monkeys who were less social, as defined by lower SEIs, showed greater vigilance to out-group monkeys. On the other hand, the most social monkey, defined by the highest SEI within the group, was equally vigilant to in-group and out-group monkeys. These findings suggest that individual variation in sensitivity to group membership will be key for understanding the complexities of social networks in both humans and animals.

Oxytocin is fundamental to parturition in females and is released during lactation to promote bonding between mothers and infants. This peptide hormone also plays a crucial role in shaping social development and promoting prosocial behavior (*Bauman et al., 2018*; *Putnam et al., 2016*). Oxytocin also impacts social behavior in males by influencing intra-group affinity, coordination, trust, and empathy (*De Dreu et al., 2011*; *Burkhart et al., 2022*; *Baumgartner et al., 2008*). Both male and female chimpanzees have shown increased levels of oxytocin before and after intergroup conflicts (*Samuni et al., 2017*). Recent evidence suggests that oxytocin influences in-group and out-group behavioral dynamics (*De Dreu, 2012*; *De Dreu and Kret, 2016*; *De Dreu et al., 2010*). Specifically, oxytocin enhances empathy and cooperation towards in-group conspecifics but increases competitiveness and hostility towards individuals from other groups (*De Dreu and Kret, 2016*; *Zhang et al., 2019*; *Shamay-Tsoory and Abu-Akel, 2016*). Our study demonstrates that oxytocin administration increases attention towards out-group individuals, while simultaneously blunting vigilance towards in-group individuals (*Figure 4*), consistent with previous work (*Jiang and Platt, 2018a*). These impacts were strongest for individual variation in prosocial tendencies indexed by grooming behavior (*Figure 4f*). These findings resonate with prior work showing that individual variation in oxytocin concentration predicts social functioning in humans (*Parker et al., 2014*) and that exogenous administration of oxytocin only improves social function in children with low endogenous levels of the peptide hormone (*Parker et al., 2017*).

Primate brains evolved to navigate the demands of living in a complex social environment (*Dunbar and Shultz, 2007*; *Cheney et al., 1986*). A growing body of evidence indicates both brain structure and function reflect social behavioral complexity (*Sallet et al., 2011*; *Testard et al., 2022*; *Sliwa and Freiwald, 2017*). For example, the so-called 'social brain network' varies in physical size as a function of both social network size and social status (*Testard et al., 2022*; *Kanai et al., 2012*; *Noonan et al., 2014*). Such structural variation may both reflect and impact biological success. For example, rhesus monkeys who become more tolerant and friendly, thereby expanding their social networks, are more likely to survive natural disasters and subsequently degraded environments (*Testard et al., 2024*). Our study unveiled subtle variation in social information processing as a function of both general social

tendencies and specific dyadic social relationships, which we hypothesize arise from variation in the structure and function of components of the social brain network.

We further hypothesize that oxytocin serves as a gain control mechanism that regulates social information processing thereby shaping multi-dimensional social relationships via distinct but interacting neural pathways (*Froemke and Young, 2021*; *Arakawa, 2021*). OT modulation of meso-cortico-limbic circuitry facilitates approach-oriented behaviors related to affiliation, familiarity, and trust, particularly towards groupmates (*De Dreu, 2012*). Oxytocin also modulates cortico-amygdala circuitry (*Striepens et al., 2012*), resulting in anxiolysis and reductions in fear-motivated action towards groupmates (*De Dreu et al., 2015*), but heightened vigilance and defensive aggression towards out-group conspecifics (*De Dreu and Kret, 2016*; *De Dreu et al., 2015*; *Harari-Dahan and Bernstein, 2014*). Our findings are consistent with this body of evidence and help to elucidate how social perception and attention vary with and potentially support multi-dimensional social relationships (*Figure 4f, k* and *Figure 5e-h*). The diverse impacts found here might be linked to many brain pathways, encompassing both the meso-cortico-limbic and cortico-amygdala circuitries. Our findings thus help to elucidate the links between the dynamics of multi-faceted social networks and the dynamics of visual attention through neuromodulatory gain control mechanisms. However, these conclusions may be constrained by the relatively small sample size and the homogeneity of a stimulus set in the study. Future research focusing on larger, more diverse cohorts and incorporating a broader range of stimuli will enhance the generalizability and applicability of the findings.

## Materials and methods

### Animal preparation

Four adult male rhesus macaques (*Macaca mulatta,* 8–13 y old, 10–12 kg) served as subjects in this study. Animals were cared for in accordance with *Guide for the Care and Use of Laboratory Animals* established by the Society for Neuroscience. All experimental procedures were approved in advance by the *Institutional Animal Administration Committees* at the Institute of Biophysics, Chinese Academy of Sciences (IBP-NHP-002(22)). Prior to experiments, we implanted a head holder to keep head stable during experiments, and sutured an eye coil to the sclera to monitor eye position by using the magnetic search coil technique (Crist Instruments, Bethesda, USA), as described previously (*Dou et al., 2023*). Monkeys were well trained to sit calmly in primate chairs (Crist Instruments, Bethesda, USA), and performed eye movement tasks in a quiet, dimly lit room. On the experimental day, all monkeys were simultaneously transported from the colony room into individual experimental rooms and participated in their tasks for about 2 hr in the morning. Then they were back to home cages and fed supplements regularly.

### Social interaction dataset

All four monkeys who participated in the study were housed in the same colony room for more than 4 y. The monkeys lived in interconnected cages that permitted direct physical interaction but also allowed to isolated them by a sliding system of partitions for animal safety. Monkey interactive behaviors were recorded by six cameras (Xiaomi PTZ version SE+, China, sample rate: 20 frames per second) simultaneously in the afternoon to evaluate their social interactions. We have collected about 560 hr video data totally, with monitoring interactive behaviors in each dyad among four subjects through the whole study.

### Automatic social interaction evaluation

YOLO or You Only Look Once is a one-stage target detection algorithm and can directly return the image pixel regression box coordinates and probability-like (*Redmon et al., 2016*). To quantify monkeys' interactive behaviors precisely and equitably, we first applied the architecture and loss function of YOLOv5 (https://github.com/ultralytics/yolov5; copy archived at *Jocher, 2025*) . We used the LabelMe tool to translate the video data into image data. We randomly chose 396 images from four monkeys as the training dataset, and labeled them according to monkeys' different behaviors: postures (upright, sitting or lying), head orientation, and rump orientation. The YOLOv5 model was used to extract the monkeys' behavior by detecting frame by frame. After training, we checked the evaluation indicators of this model, including loss value of YOLOv5 in the training process, changes of

mAP, confusion matrix. and PR curve. And, we found the YOLOv5 worked effectively on the dataset with mean average precision (IoU = 0.5) of 0.97 (*Figure 1—figure supplement 1*). The behavior performances of each subject in any given dyad were identified as three dimensions of social behaviors (*Figure 1d–f*): proximity (approach more-approach less), grooming (groomee-groomer), aggression (aggressor-victim). To ensure accuracy, we also had two observers—one animal caretaker and one lab member, both blind to the research purposes — independently mark monkey social behaviors from representative videos for all possible monkey dyads. Their identifications of behavior aligned well with the automatic detections.

Then, we graded clearly delineated antagonistic engagements in a 1 and –1 dichotomous fashion. In any dyad of interaction sessions, one monkey was deemed score 1 to the other if their actions occurred more times in one direction than in the other, such as showing more proximity, receiving more grooming, and initiating a physical aggression more frequently, while the other one was scored –1. Aggressive behaviors unfold with remarkable swiftness, with reciprocal responses occurring on a timescale measured in seconds. Therefore, we initially employed YOLOv5 for temporal assessment, followed by manual determination of the initiation sequence of attacks. Subsequently, we meticulously recorded the durations during which each party actively initiated aggression, thereby discerning the victor and vanquished in instances of aggressive behavior. If there were no related social interactions happened, both of them gained zero as their interaction scores.

## Visual orienting task

Visual stimuli were presented at a refresh rate of 100 Hz on a monitor (VG278Q ASUS, China) positioned 40 cm from the monkey. The screen was subtended with a visual field of 73 × 45° with a 1920 × 1080 resolution. A computer running laboratory's Maestro software controlled experimental stimuli and task timing. In the daily experiments, a 0.5° white square was served as a target. A trial began with the target located in the center of the black screen (RGB: 0, 0, 0) for a variable period of 800–1200 ms to minimize anticipatory movements, and monkeys were required to fixate within an invisible 2° × 2° window. Every 50 trials were served as a session including forty-eight social interference trials (96%) and two visuo-motor trials (4%) for data evaluation (*Figure 2a*). In the social interference task, a 7° circled picture was randomly presented 18° eccentric either on the right or left and briefly flashed as a distractor for 100ms. After the first 50 ms, the white target relocated 14° from center point on one of two locations: congruent (same hemifield with distractor) or incongruent (opposite hemifield), to guide saccadic eye movements. In the visual-motor task, only the target was randomly presented on one of hemifield without a distractor interference in order to assess monkeys' behavioral performances. All social interference and visual-motor trials were introduced in a random order and one time in a session. All sessions were continuously run in an experimental day without pause. Thus, monkeys could be incapable to detect the testing session. Once monkeys made a correct saccade to the target within an interval of 300 ms from the target relocation, a few droplets of liquid were delivered as reward.

Distractors were composed of seven face pictures from different monkeys and their scrambled images and adjusted their luminance to 54.67±2.69 cd/m². Four of them were taken from subjects participated in the study, who had social interactions in the same colony room. The other three monkey face pictures were sampled from outside the social group (*Figure 2b*). For each trial, subject was introduced a distractor randomly chosen from face pictures of his three social interacted groupmates, or the three out-group monkeys without social interactions, or the related six scrambled images (randomly shuffled the pixels), with equal chances. All face pictures were well controlled with a neutral expression and forward gaze, and subtended by a 7° circle. Prior to this study, all these distractors have been daily introduced to four subjects as a target to evoke visual guided saccadic eye movements for 8 wk. Thus, subjects were well exposed to all monkey face pictures and the scrambled images, no matter having social interactions with the monkey back to home cages, or not.

## Pharmacological manipulation

Prior to the pharmacological treatment, all monkeys had become habituated to the nebulizer-assisted aerosolized exposure procedure and had been receiving saline delivery for around 2 wk. On the alternating experimental days, oxytocin (40 IU/mL, 2 mL; Beyotime) and saline (2 ml, served as control) were randomly delivered via nebulization (Omron, baby nebulizer) into monkey's nose and mouth

continuously for 2 min (*Bauman et al., 2018*). After 15 min relax in the dim room, monkeys participated in the visual orienting task.

## Data acquisition

We measured eye position signals using a magnetic search coil system (Crist Instruments, Bethesda, Maryland, United States) to estimate eye movements. After passing the signals through an analog differentiator, voltages proportional to horizontal and vertical eye velocity were generated. The differentiator contained a filter that rejected signals above 25 Hz (–20 dB per decade) and produced eye movement signals at lower frequencies. Eye signals were sampled at 1 kHz and stored for analysis offline. Eye traces in each trial were inspected by a custom-built MATLAB program and then manually reviewed. The identification of saccadic eye movements was based on velocity and acceleration thresholds (velocity, 50°/s; acceleration, 500°/s$^2$). If saccade occurred during the 500ms fixation interval preceding the onset of the distractor, the trial was discarded. Saccadic eye movements were quantified in terms of the response time (the latency of saccade to the target) and the initial eye movement direction profile.

## Statistical analysis

### Drifting time and distractor bias

To evaluate the effects of the monkey face on saccadic eye movements, we calculated the drifting time (DT) between the saccadic latencies when the distractor was a monkey face picture $(RT_f)$ or its scrambled image $(RT_s)$ in a certain visual condition (congruent or incongruent) on each experimental session:

$$DT = RT_f - RT_s \tag{1}$$

Furthermore, we found that our data were consistent with prior research: the congruent distractor accelerated the saccadic eye movement, whereas the incongruent distractor delayed saccade initiation. To summarize the face effects on monkeys' behavioral performances across two conditions, we defined the distractor bias to evaluate the influence:

$$Distractor\,Bias = DT_{Incon} - DT_{Con} \tag{2}$$

where $DT_{Incon}$ and $DT_{Con}$ are the mean drifting times under incongruent and congruent conditions on each experimental day. This use of eye responses evoked by various monkey face pictures and their scrambled images under two conditions assigned the complete effect of the feature of face, with well control.

To assess the effect of distractors on the initial choice of gaze shifting, we calculated the probability of the first saccade being directed towards the distractor in the incongruent condition. We then computed the difference in the probabilities that would be directed towards intact monkey faces and scrambled images to evaluate the impact of face information on attention.

### Social engagement index

We utilized linear models to combine the three dimensions of social relationships (aggression, grooming, and proximity). SEI summarizes the social interactions between a specific monkey and the other three monkeys in the group, which could provide an assessement on general social tendencies (*Figure 3a*). We modeled the social engagement index as:

$$SEI = \omega_a \times \sum_{i=1}^{n} A_i + \omega_g \times \sum_{i=1}^{n} G_i + \omega_p \times \sum_{i=1}^{n} P_i \tag{3}$$

where $\omega_a$, $\omega_g$, and $\omega_p$ indicate the weights of three dimensions of social behaviors separately. The $\omega$ constrains between 0 and 1 with a step of 0.02. $i$ presents the $n^{th}$ interact partner for each individual. In our dataset, $n$ presents 3 socially interacting monkeys in the colony group. $A$, $G$, and $P$ are interaction scores for aggression, grooming, and proximity.

### Individual engagement index

IEI evaluates the social interactions between a given subject and his specific counterpart, which could offer a perspective at the specific individual dyadic level (*Figure 5a*). The IEI was calculated as:

$$IEI_{A(B)} = \omega_a \times A_{A(B)} + \omega_g \times G_{A(B)} + \omega_p \times P_{A(B)} \tag{4}$$

where $\omega_a$, $\omega_g$, and $\omega_p$ again indicate the independent weights for three behavioral dimensions separately. We adopted identical ranges and steps for $\omega$, constraining it between 0 and 1, with a step of 0.02. $IEI_{A(B)}$ indicates the interactions within a dyadic interacted pair, actor Monkey A towards his partner Monkey B.

Statistical analyses were conducted to examine the significance of the results across four monkeys and experimental days, with the significance level at 0.05. Two-sided Pearson's correlation coefficient was utilized to quantify the correlations. If not stated otherwise, data were analyzed using a two-tailed paired Student's t-test.

## Acknowledgements

We thank members of our laboratory for their helpful comments on an earlier version of the manuscript and for helpful discussions. Chen Wu, Qian Wang, and Yan-hui Fu at the Institute of Biophysics, as well as Stephen G Lisberger, and Scott Ruffner, for their invaluable technical assistance. STI2030-Major Projects 2022ZD0204800 (YY) Beijing Natural Science Foundation Z210009 (YY) National Natural Science Foundation of China 32070987 (YY) National Natural Science Foundation of China 31722025 (YY)

## Additional information

### Funding

| Funder | Grant reference number | Author |
|---|---|---|
| STI2030-Major Projects | 2022ZD0204800 | Yan Yang |
| Beijing Natural Science Foundation | Z210009 | Yan Yang |
| National Natural Science Foundation of China | 32070987 | Yan Yang |
| National Natural Science Foundation of China | 31722025 | Yan Yang |

The funders had no role in study design, data collection and interpretation, or the decision to submit the work for publication.

### Author contributions

Sainan Liu, Conceptualization, Data curation, Formal analysis, Validation, Investigation, Visualization, Methodology, Writing – review and editing; Jiepin Huang, Data curation, Investigation, Writing – review and editing; Suhao Chen, Data curation, Software, Formal analysis, Investigation, Methodology; Michael L Platt, Investigation, Methodology, Writing – review and editing; Yan Yang, Conceptualization, Data curation, Supervision, Funding acquisition, Validation, Investigation, Visualization, Methodology, Writing – original draft, Project administration, Writing – review and editing

### Author ORCIDs

Sainan Liu ⓘ https://orcid.org/0009-0001-8422-6503
Suhao Chen ⓘ http://orcid.org/0009-0005-1496-7858
Yan Yang ⓘ https://orcid.org/0000-0003-3001-9178

### Ethics

Animals were cared for in accordance with Guide for the Care and Use of Laboratory Animals established by the Society for Neuroscience. All experimental procedures were approved in advance by the

Institutional Animal Administration Committees at the Institute of Biophysics, Chinese Academy of Sciences (IBP-NHP-002(22)).

Reviewer #1 (Public review): https://doi.org/10.7554/eLife.104460.3.sa1
Reviewer #2 (Public review): https://doi.org/10.7554/eLife.104460.3.sa2
Author response https://doi.org/10.7554/eLife.104460.3.sa3

---

## Additional files

### Supplementary files
MDAR checklist

### Data availability
All data generated or analyzed during this study are included in the manuscript and supporting figures.

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
