## [Editor Report · eLife Assessment]

This study examined how multidimensional social relationships influence social attention in rhesus macaques, linking individual and group-level behaviors to attentional processes. The findings that oxytocin altered social attention and its relationship to both social tendencies and dyadic relationships are **important**, as recent technological advances allow for the exploration of neuronal activities and mechanisms in free-moving macaques. This work is **convincing** and will be of interest to those studying the interplay between social dynamics and information processing in primates.

---

## [Referee Report · Reviewer #1 (Public review)]

Summary:

This study aims to investigate the links between social behaviors observed in free-moving situations and behavioral performances measured in well-controlled, laboratory settings. The authors assessed general social tendencies and dyadic relationships among four monkeys in a group by scoring agonistic (aggression) and affiliative (grooming and proximity) behaviors in each pair. By measuring the saccadic reaction time in a classic social interference task, the authors reported that the monkeys with higher SEIs (i.e., more social individuals) were less distracted by the faces of other monkeys. These effects were enhanced when the distractors were out-group monkey faces rather than in-group ones. Lastly, oxytocin administration increased the impact of the out-group monkey faces in the social interference task, while reducing the magnitude of general social tendencies measured with SEI.

Strengths:

(1) The combination of behavioral data obtained in a colony room and in a laboratory environment is rare and important.

(2) The evaluation of social interactions were successfully performed based on an automated target detection algorithm. The resulting multi-dimensional, complicated social interactions were summarized into simple indices (SEI and IEI). These indices provide a good measure for the social tendencies of each monkey.

(3) Well-designed and robust experiments in the laboratory environment that are linked nicely with the general social tendencies observed in spontaneous behaviors.

Weaknesses:

(1) While the overall results are interesting, I am somewhat left confused about how to interpret the difference in the scores derived from different conditions. For example, the authors stated "Comparing the weights for in-group and out-group distractors, the effect of proximity was larger than that of aggression and grooming" in p.8. Does this mean that the proximity is indeed the type of behavior most affected in the out-group condition compared to the in-group condition? The out-group effects are difficult to examine with actual behavioral data, but some in-group effects such as those involving OT can be tested, which possibly provides good insights into interpreting the differences of the weights observed across the experimental conditions.

(2) I think it is important to provide how variable spontaneous social interactions were across sessions and how impactful the variability of the interactions is on the SEI and IEI, as it helps to understand how meaningful the differences of weights are across the conditions, but such data are missing. In line with this point, although the conclusions still hold as those data were obtained during the same experimental periods, shouldn't the weights in Fig. 3f and Figs. 4g and 4h (saline) be expected to be similar, if not the same?

Comments on revisions: I do not have further comments.

---

## [Referee Report · Reviewer #2 (Public review)]

Summary:

The study presents significant findings that elucidate the relationship between multi-dimensional social relationships and social attention in rhesus macaques. By integrating advanced computational methods, behavioral analyses, and neuroendocrine manipulation, the authors provide strong evidence for how oxytocin modulates attention within social networks. The results are robust and address critical gaps in understanding the dynamics of social attention in primates.

Strengths:

(1) The use of YOLOv5 for automatic behavioral detection is an exceptional methodological advance. The combination of automated analyses with manual validation enhances confidence in the data.

(2) The study's focus on three distinct dimensions of social interaction (aggression, grooming, and proximity) is comprehensive and provides nuanced insights into the complexity of primate social networks.

(3) The investigation of oxytocin's role adds a compelling neuroendocrine dimension to the findings, providing a bridge between behavioral and neural mechanisms.

Weaknesses:

(1) The study's conclusions are based on observations of only four monkeys, which limits the generalizability of the findings. Larger sample sizes could strengthen the validity of the results.

(2) The limited set of stimulus images (in-group and out-group faces) may introduce unintended biases. This could be addressed by increasing the diversity of stimuli or incorporating a broader range of out-group members.

Comments on revisions: I have no further comments!

---

## [Author Response]

The following is the authors’ response to the original reviews.

Reviewer #1 (Public review):Weaknesses:(1) While the overall results are interesting, I am somewhat left confused about how to interpret the difference in the scores derived from different conditions. For example, the authors stated "Comparing the weights for in-group and out-group distractors, the effect of proximity was larger than that of aggression and grooming" in p.8. Does this mean that the proximity is indeed the type of behavior most affected in the out-group condition compared to the in-group condition? The out-group effects are difficult to examine with actual behavioral data, but some in-group effects such as those involving OT can be tested, which possibly provides good insights into interpreting the differences of the weights observed across the experimental conditions.

Thank you for your thoughtful comments and for highlighting an important aspect of our findings. The statement in page 8 refers to the relative impact of different social behaviors—proximity, aggression, and grooming—on the derived weights for in-group and out-group distractors. Specifically, the data suggest that proximity exerts a stronger influence than aggression or grooming in differentiating the effects of out-group versus in-group distractors. Regarding the out-group condition, we acknowledge that it presents challenges for direct behavioral observation, as interactions involving out-group members are often more difficult to quantify in naturalistic settings. However, we agree with you about the suggestion to test certain in-group effects, particularly those influenced by oxytocin (OT), as they offer a more controlled framework to validate and interpret the observed differences in weights across experimental conditions. In line with this, we examined specific in-group behaviors under OT administration to disentangle their contributions to attentional dynamics (Fig. 4 and Fig. 5 e to h). By integrating controlled experimental manipulations, we think these results could provide deeper insights into how social relationships shape the observed patterns of attention.

(2) I think it is important to provide how variable spontaneous social interactions were across sessions and how impactful the variability of the interactions is on the SEI and IEI, as it helps to understand how meaningful the differences of weights are across the conditions, but such data are missing. In line with this point, although the conclusions still hold as those data were obtained during the same experimental periods, shouldn't the weights in Fig. 3f and Figs. 4g and 4h (saline) be expected to be similar, if not the same?

Thank you for your insightful comments. As highlighted, we utilized the entire experimental period as the dataset to evaluate the monkeys' social interactions. The experiments presented in Figures 3 and 4 were designed to examine how social relationships correlate with patterns of social attention under two distinct conditions: without manipulation (Fig. 3) and with nebulized exposure to oxytocin and saline (Fig. 4). Theoretically, the weights observed in the unmanipulated condition and the nebulized saline condition should be similar. However, our results indicate that distractor biases shifted significantly following nebulized saline exposure (Fig. 4) compared to the unmanipulated condition (Fig. 3) (MK: p = 9.3×10^-3^, ML: p = 9.77×10^-4^, MC: p = 9.77×10^-4^, MA: p = 0.09; n_1_ = n_2_ = 12 experimental days; Two-sided Wilcoxon signed-rank test). This suggests that the nebulization process itself, despite acclimating the monkeys to saline exposure for approximately two weeks prior to the experiments, still influenced their attentional behaviors.

While the primary goal of nebulization was to assess the effects of oxytocin on social attention, our main conclusions remain robust, even considering the impact of nebulization on distractor biases. We acknowledge that variability in spontaneous social interactions across days or experimental sessions could be an important factor influencing the SEI and IEI. The dynamic nature of social interactions within the colony is likely affected by numerous variables. Future research will aim to integrate these factors into a more comprehensive and dynamic framework to better interpret their influence on social attention metrics.

**Reviewer #2 (Public review):**
Weaknesses:(1) The study's conclusions are based on observations of only four monkeys, which limits the generalizability of the findings. Larger sample sizes could strengthen the validity of the results.

Thank you for your valuable comment. We acknowledge that the relatively small sample size could influence the generalizability of the findings. However, despite this limitation, our work systematically examined multifaceted social relationships among monkeys and their attentional strategies within a well-controlled experimental setup. We reported results across sessions and conditions (e.g., in-group vs. out-group; saline vs. Oxytocin), which strengthens the reliability of the observed effects of social networks within this context. We agree that increasing the sample size would improve the generalizability of the results. Future studies with a larger cohort will be critical for confirming the robustness of our findings and expanding their broader applicability. We have acknowledged this limitation in the revised manuscript and highlighted the potential for further research with larger sample sizes to validate and extend our conclusions.

(2) The limited set of stimulus images (in-group and out-group faces) may introduce unintended biases. This could be addressed by increasing the diversity of stimuli or incorporating a broader range of out-group members.

Thank you for your thoughtful comment. We acknowledge that the use of a limited set of six monkey faces as stimuli for in-group and out-group conditions could potentially introduce biases. To address this concern, we conducted an additional analysis to minimize the potential impact of individual images on our findings using the current dataset. Specifically, we randomly excluded one in-group and one out-group image and reanalyzed distractor biases using the remaining two images (Supplementary Fig. 3a). For each subject, this approach generated three sets of two distractors per group, resulting in 81(3^4^) combinations across four monkey subjects, and a total of 81 × 81 subject-distractor pairings. We statistically compared distractor biases between in-group and out-group faces for each combination (Supplementary Fig. 3b). As shown in Supplementary Fig. 3c, 99.30% of the 6,561 combinations demonstrated significantly lower distractor biases towards in-group faces compared to out-group faces (two-sided Wilcoxon signed-rank test, *p* < 0.05). These results suggest that the observed differences in social attention between in-group and out-group monkeys are unlikely to be driven by specific images within the stimulus set. That said, we agree that increasing the diversity of stimulus images or incorporating a broader range of out-group members would improve the generalizability of the results. We have acknowledged this limitation in the revised manuscript and highlighted the potential for further research to incorporate a more diverse stimulus set to validate and extend our findings.

“However, these conclusions may be constrained by the relatively small sample size and the homogeneity of stimulus set in the study. Future research focusing on larger, more diverse cohorts and incorporating a broader range of stimuli will enhance the generalizability and applicability of the findings.”

**Reviewer #1 (Recommendations for the authors):**
It is difficult to distinguish "Getting fighted" and "Fighting partner" in Fig. 1b (esp. when printed). I thought Actor showed "Fighting partner" several times in Session 2, but it seems to be "Getting fighted" judging from Figs. 1c and 1d. Is this correct? If so, I would suggest to change the color to improve visibility.

Thank you for your valuable comment. We apologize for the confusion in the previous version. To improve clarity, we have both terms to “begin fighting” and “being fought”. As shown in Figure 1b, we now explicitly define the identities of the two monkeys as the actor (K) and the partner (L), with all behaviors described from the perspective of the actor. For example, when the actor (K) initiates the fight, it is marked as “begin fighting”, whereas when the partner (L) initiates the fight, the actor (K) is the recipient and labeled as “being fought”. Additionally, we have implemented your suggestion by changing the colors to enhance visibility, especially for the terms “begin fighting” and “being fought”.

**Reviewer #2 (Recommendations for the authors):**
I have some minor concerns:(1) Figure1B, caption for x axis is missing, 4 means 4 days?

Thank you so much for the comment. We have clarified the x-axis in Figure 1B, where the label "4" corresponds to 4 hours of video typing on each experimental day. The revised figure now includes the appropriate label for better clarity. We appreciate your careful attention to this detail.

(2) I am slightly concerned about animal safety. How do the experimenters ensure the animals' safety and well-being in cases of aggressive interactions or attacks?

Thank you for your comment. We share your concern regarding animal safety and take re the well-being of the monkeys in the study. All experimental procedures were reviewed and approved by the Institutional Animal Care and Use Committee at the Institute of Biophysics, Chinese Academy of Sciences (IBP-NHP-002(22)). The monkeys were housed together in the same colony room for over four years, in interconnected cages that allowed for direct physical interaction. Animal behaviors in cages were closely monitored via a live video system to ensure their safety. To prevent potential injuries, a sliding partition system was in place, enabling the isolation of individual animals when necessary, minimizing risks to their well-being.